# GEMFLOW: DYNAMIC GRAPH EVOLUTION-AWARE MASKED PRE-TRAINING FOR TRAFFIC FLOW FORECASTING

## ABSTRACT

Traffic flow forecasting is inherently challenging due to the continuous evolution of spatial dependencies and the coexistence of heterogeneous temporal patterns. Most existing pre-training methods either rely on static graphs or employ generic masking strategies that overlook the dynamic nature of road networks, limiting their robustness and transferability. To overcome these limitations, we propose GEMFlow (Graph Evolution-aware Masking for traffic Flow forecasting), a novel pre-training framework that unifies masked representation learning with adaptive graph evolution modeling. Specifically, GEMFlow introduces a curriculum-style dynamic masking strategy that operates on temporal patches while conditioning the masking process on the evolution of graph structures. This design allows the model to emphasize informative temporal segments and adapt to structural drift across time, going beyond prior decoupled masking approaches. The learned graph evolution-aware representations can be seamlessly transferred to diverse downstream forecasting models without modifying their architectures. Extensive experiments on four real-world PeMS datasets demonstrate that GEMFlow achieves state-of-the-art performance, consistently improving accuracy, efficiency, and robustness. Moreover, qualitative analysis of the learned dynamic graphs reveals interpretable evolution patterns, highlighting the potential of GEMFlow as a versatile pre-training paradigm for spatiotemporal forecasting.

## 1 INTRODUCTION

Traffic flow is a typical form of spatiotemporal data, capturing how vehicle movements evolve across road networks over time. As a fundamental element of intelligent transportation systems, traffic flow analysis plays a vital role in applications such as congestion management, route planning, and urban mobility optimization. Similar to other spatiotemporal phenomena—such as weather dynamics or epidemic spread—traffic flow exhibits complex spatial interactions and temporal variations that demand careful modeling. Effective analysis of traffic flow can uncover underlying dynamics, support accurate forecasting, and ultimately enable data-driven decision-making for resource allocation, infrastructure planning, and sustainable urban development.

Traffic flow inherently reflects dependencies across both spatial and temporal dimensions, yet these relationships are rarely stable. A central challenge is **spatiotemporal inconsistency**, where spatially or temporally adjacent observations exhibit shifting or irregular correlations. As illustrated in Figure 1(a), the traffic sensors located at A (Sensor 20) and C (Sensor 301) display similar flow patterns during Period 1 (gray region in Figure 1(b)), primarily due to their geographic proximity. However, in Period 2 (yellow region), Sensor 20 becomes more correlated with Sensor 23 at Location B, while its similarity with Sensor 301 weakens. This dynamic reconfiguration highlights how external factors—such as variations in travel demand, road conditions, or infrastructure adjustments—can alter correlation structures over time, underscoring the necessity of modeling graph evolution rather than assuming static spatial dependencies.relationships over time.

Another fundamental property of traffic flow is its **multiscale nature**, where patterns emerge at multiple temporal resolutions. In traffic flow analysis, this manifests in both fine-grained variations (e.g., hourly rush hours) and coarse-grained periodicities (e.g., daily or weekly cycles). As shown

in Figure 1(c), sensor readings on January 3rd and January 31st capture not only consistent short-term fluctuations but also evolving long-term trends. The shaded regions highlight peak (gray) and trough (yellow) periods, demonstrating how local dynamics persist while global patterns shift due to seasonal changes, infrastructure updates, or collective behavioral adaptation. These hierarchical structures are widely present across domains—from climate modeling to urban mobility—and must be effectively captured to achieve accurate forecasting, robust anomaly detection, and informed decision-making in intelligent systems.

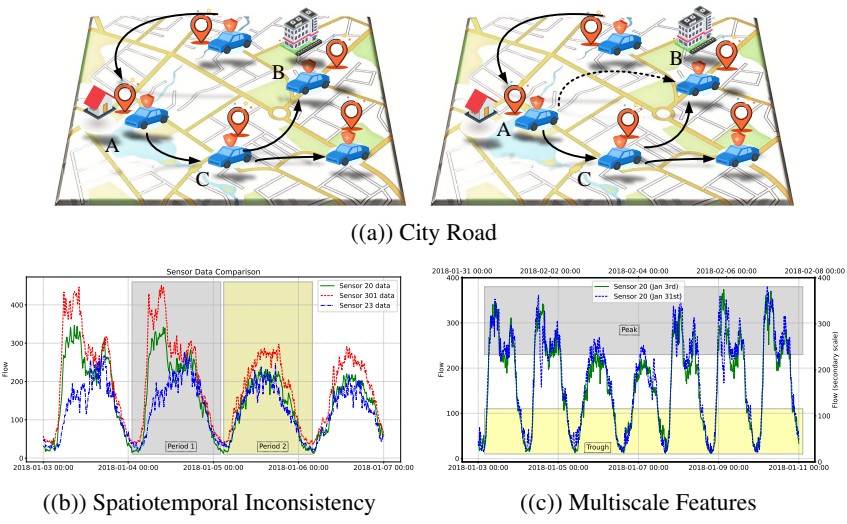

((a)) City Road

((b)) Spatiotemporal Inconsistency      ((c)) Multiscale Features

Figure 1: Examples of dynamic traffic flow in city

Previous research has explored various approaches to traffic flow forecasting. Statistical models Fang et al. (2021); Song et al. (2020) assume linearity and stationarity, offering interpretable yet limited frameworks for dynamic traffic systems. Hybrid architectures such as GCN+RNN Li et al. (2018) capture spatial dependencies through graph convolutions and temporal patterns via recurrent units, while GCN+TCN methods Yu et al. (2018); Wu et al. (2019); Guo et al. (2019) enhance temporal modeling using convolutional structures. More recently, GNN-based pre-training techniques Gao et al. (2024) have been proposed to learn transferable traffic representations from large-scale datasets, improving downstream forecasting performance. Despite these advancements, existing methods still struggle with the complex nature of traffic flow dynamics. Temporal correlations are often inadequately modeled, particularly across varying time intervals such as peak and off-peak hours. Spatial relationships—such as the evolving influence of nearby road segments or sensors—are frequently underutilized. Moreover, capturing spatiotemporal interactions remains challenging, leading to oversimplified representations of traffic dynamics. Finally, most models fail to simultaneously account for fine-grained local fluctuations and long-term global trends, thereby limiting both predictive accuracy and generalizability.

In this study, we propose GEMFlow, a pre-training framework for traffic flow forecasting that captures dynamic spatial interactions and multi-scale temporal patterns. Inspired by masked pre-training Vaswani et al. (2017), GEMFlow reconstructs masked traffic segments under evolving graph structures, enabling robustness to structural drift and adaptive modeling of traffic dependencies. The learned representations transfer seamlessly to diverse forecasting backbones, improving accuracy, efficiency, and interpretability. Our contributions are:

- We design **GEMFlow**, a plug-and-play pre-training framework that learns transferable traffic representations, enhancing diverse forecasting models without architectural changes.
- We introduce a **graph evolution-aware masking strategy**, which effectively models dynamic spatial dependencies and multi-scale temporal patterns in traffic networks.
- We validate GEMFlow on four real-world PeMS datasets, demonstrating consistent improvements over strong baselines and uncovering interpretable patterns of traffic graph evolution.

## 2 RELATED WORK

### 2.1 TRAFFIC FLOW FORECASTING

Traffic flow forecasting aims to predict future traffic states by leveraging historical observations and road network structures Gao et al. (2024). Early approaches based on recurrent and convolutional networks captured temporal dynamics and local spatial features but struggled with long-range dependencies and complex non-Euclidean structures. To address these challenges, Graph Neural Networks (GNNs) were introduced Yu et al. (2018), leading to a series of Spatio-Temporal Graph Neural Networks (STGNNs) Wu et al. (2019); Han et al. (2021); Jiang et al. (2023b) that significantly improved the modeling of spatial–temporal correlations. Building on this foundation, recent works have advanced along multiple directions: decomposition-based models such as STDN Cao et al. (2025), D$^2$STGNN Jiang et al. (2023b), and STWave Fang et al. (2023) disentangle traffic signals into components like trend, seasonal, diffusion, inherent, or event patterns to better capture dynamics; node- and heterogeneity-aware approaches including STPGNN Kong et al. (2024) and HimNet Dong et al. (2024) emphasize pivotal nodes or adapt parameters from heterogeneity-informed meta-pools; and frequency- or dynamic graph-based methods such as DFDGCN Li et al. (2024) leverage Fourier transform, identity/time embeddings, and hybrid graph structures to enhance robustness. Despite these advances, most existing models remain constrained by short input horizons, static or simplified spatial graphs, and limited capacity to capture evolving correlations, which motivates the development of new pre-training paradigms that can learn transferable traffic representations while explicitly modeling graph evolution and multi-scale temporal dynamics.

### 2.2 SPATIOTEMPORAL-AWARE MASKED PRE-TRAINING

Masked pre-training has proven highly effective as a self-supervised strategy in natural language processing (NLP) and computer vision (CV). In NLP, models such as BERT Devlin et al. (2019) and ALBERT Lan et al. (2020) learn contextualized representations by predicting masked tokens from bidirectional context. In CV, methods like BEiT Bao et al. (2022) and MAE He et al. (2022) reconstruct masked image patches to capture meaningful visual features. Motivated by these successes, researchers have recently extended masked pre-training to time series forecasting Shao et al. (2022); Li et al. (2023); Gao et al. (2024). Nevertheless, most existing approaches either ignore spatial dependencies or process channels independently, which limits their ability to model complex spatiotemporal correlations. Meanwhile, alternative directions such as BigST Han et al. (2024) focus on scalability with linear-complexity architectures, but do not exploit masked pre-training to capture fine-grained spatiotemporal heterogeneity. We propose **GEMFlow**, a graph-enhanced masked pre-training framework that learns long-range spatiotemporal dependencies and dynamic correlations in traffic networks, offering plug-and-play transfer to diverse forecasting models.

## 3 PRELIMINARIES

**Spatiotemporal Traffic Data.** Traffic flow forecasting is typically based on spatiotemporal data collected from sensor networks deployed across road systems. Formally, the data is denoted as $\mathbf{X} \in \mathbb{R}^{N \times T \times F}$, where $N$ is the number of sensors (nodes), $T$ is the number of time steps, and $F$ is the feature dimension (e.g., traffic speed, flow, or occupancy). Each entry $\mathbf{X}_{i,t,:}$ records the traffic state of sensor $i$ at time step $t$.

**Graph Representation and Evolution.** The spatial structure of the road network is commonly modeled as a graph $\mathcal{G}_t = (\mathcal{V}, \mathcal{E}_t, \mathbf{A}_t)$, where $\mathcal{V}$ is the set of $N$ sensors, $\mathcal{E}_t$ is the set of dynamic edges at time $t$, and $\mathbf{A}_t \in \mathbb{R}^{N \times N}$ is the adjacency matrix encoding spatial dependencies. Unlike static graphs, real-world traffic networks evolve over time due to factors such as congestion, road conditions, or external interventions, making *graph evolution modeling* essential for accurate forecasting.

**Pre-training Paradigm.** To capture transferable spatiotemporal representations, GEMFlow adopts a masked autoencoding (MAE) objective. Given a subsequence $\mathbf{X}_{t:t+k-1}$ of length $k$, a binary mask $\mathbf{M}$ is applied to hide part of the data. The encoder $f(\cdot)$ is trained to reconstruct the masked information from the visible context:

$$\mathcal{L}_{pre} = \mathcal{L}\big((1-\mathbf{M}) \odot \mathbf{X}_{t:t+k-1}, \, f(\mathbf{M} \odot \mathbf{X}_{t:t+k-1}, \mathbf{A}_{t:t+k-1})\big), \tag{1}$$

where $\mathbf{A}_{t:t+k-1}$ provides the evolving graph structure within the pre-training window. This enables the encoder to learn both temporal dependencies and dynamic spatial correlations.

**Forecasting Paradigm.** In the downstream stage, given past $T_h$ steps $\mathbf{X} \in \mathbb{R}^{N \times T_h \times F}$, the model predicts the next $T_f$ steps $\mathbf{Y} \in \mathbb{R}^{N \times T_f \times C}$:

$$\mathcal{L}_{fore} = \mathcal{L}\big(\mathbf{Y}, \, g(f(\mathbf{X}_{t:t+k-1}, \mathbf{A}_{t:t+k-1}), \mathbf{X}_{T_h})\big), \tag{2}$$

where $f(\cdot)$ is the pre-trained encoder and $g(\cdot)$ is the forecasting head. This paradigm bridges dynamic graph evolution-aware masked pre-training with traffic flow forecasting, ensuring both scalability and transferability.

## 4 METHODOLOGY

This section delves into the technical specifics of our GEMFlow, as delineated in Figure 2.

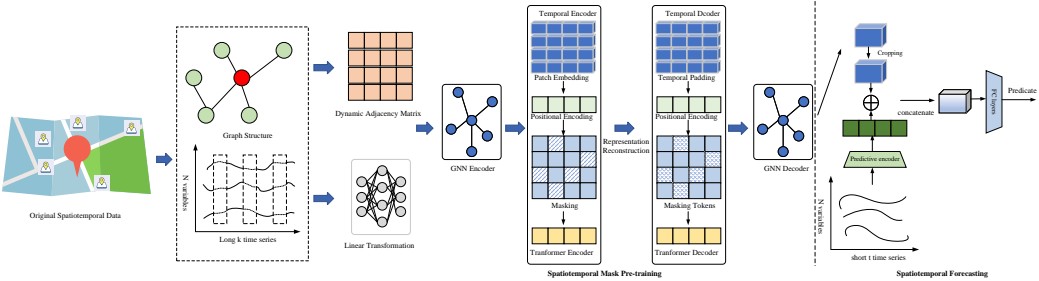

Figure 2: The framework of GEMFlow.

### 4.1 DYNAMIC GRAPH EVOLUTION MODELING

**Parameterized dynamic adjacency.** Traffic correlations are inherently time-varying: static adjacency matrices predefined by distance cannot capture fluctuations caused by demand shifts, road conditions, or external events. We therefore parameterize the adjacency and allow it to evolve over time, so that $\hat{\mathbf{A}}_t$ can adaptively reflect dynamic correlations. This design combines node similarity (capturing flexible data-driven relationships), temporal context (modeling lag-dependent effects such as decay and periodicity), and structural priors (preserving stable long-term connectivity). Such a hybrid formulation yields adjacency matrices that are adaptive yet regularized, enabling robust modeling of evolving spatial dependencies.

$$\mathbf{S}_t = \frac{\phi(\mathbf{U}_t)\phi(\mathbf{U}_t)^\top}{\sqrt{d}} + \psi(\Delta t) + \eta \mathbf{A}^{\text{static}}, \tag{3}$$

where the three terms correspond to complementary components: (1) $\frac{\phi(\mathbf{U}_t)\phi(\mathbf{U}_t)^\top}{\sqrt{d}}$ is a similarity-based score that captures pairwise relationships between node embeddings at time $t$. Here, $\mathbf{U}_t \in \mathbb{R}^{N \times d}$ denotes the temporal node embeddings, $\phi(\cdot)$ is a learnable projection, and the scaling factor $\sqrt{d}$ stabilizes attention scores. (2) $\psi(\Delta t)$ encodes temporal bias as a function of the time lag $\Delta t$, incorporating effects such as decay and periodicity. This term modulates edge strengths according to temporal distance. (3) $\eta \mathbf{A}^{\text{static}}$ introduces a static prior adjacency matrix that reflects long-term invariant connectivity patterns (e.g., geographical distance or predefined road network links), with $\eta$ controlling its relative importance. The final dynamic adjacency matrix is obtained by applying a non-linear transformation and normalization to $\mathbf{S}_t$, followed by sparse selection:

$$\hat{\mathbf{A}}_t = \text{Top-}k\left(\text{RowSoftMax}\left(\sigma(\mathbf{S}_t)\right)\right), \tag{4}$$

where $\sigma(\cdot)$ denotes an activation function (e.g., ReLU), $\text{RowSoftMax}(\cdot)$ normalizes each row into a probability distribution, and $\text{Top-}k(\cdot)$ retains only the $k$ strongest connections per node to ensure sparsity and interpretability.

**Temporal Encoding via Causal Transformer.** To capture long-range temporal dependencies without information leakage, we employ a causal Transformer encoder over a sliding historical window $\mathbf{X}_{t-w+1:t} \in \mathbb{R}^{N \times w \times C}$.

First, node-wise temporal tokens are obtained via linear projection and positional encoding:

$$\mathbf{Z}_t = \text{Proj}(\mathbf{X}_{t-w+1:t}) + \mathbf{E}_{\text{time}} \quad \in \mathbb{R}^{N \times w \times d}, \tag{5}$$

where $\mathbf{E}_{\text{time}}$ is a learnable temporal positional encoding.

Then, causal self-attention is applied along the temporal dimension:

$$\text{Attn}(\mathbf{Q}, \mathbf{K}, \mathbf{V}) = \text{softmax}\left(\frac{\mathbf{Q}\mathbf{K}^\top}{\sqrt{d}} + \mathbf{M}_{\text{causal}}\right) \mathbf{V}, \tag{6}$$

where $\mathbf{Q} = \mathbf{Z}_t \mathbf{W}_Q$, $\mathbf{K} = \mathbf{Z}_t \mathbf{W}_K$, $\mathbf{V} = \mathbf{Z}_t \mathbf{W}_V$ are the query, key and value matrices, and $\mathbf{M}_{\text{causal}}(i,j) = -\infty$ if $j > i$ ensures that each step $i$ only attends to its past and current positions.

The temporal encoder output is

$$\mathbf{H}_t = \text{TransformerEnc}_{\text{causal}}(\mathbf{Z}_t) \in \mathbb{R}^{N \times w \times d}, \tag{7}$$

and the embedding for the current step $t$ is obtained by selecting the last hidden state:

$$\mathbf{U}_t = \mathbf{H}_t[:, w, :] \quad \in \mathbb{R}^{N \times d}. \tag{8}$$

**Temporal bias.** We modulate temporal distances with a lightweight yet expressive bias:

$$\psi(\Delta t) = w_{b(\Delta t)} + \alpha\, e^{-\beta\, \Delta t} + \sum_{p \in \mathcal{P}} \left[ a_p \cos\left(\frac{2\pi\Delta t}{p}\right) + b_p \sin\left(\frac{2\pi\Delta t}{p}\right) \right], \tag{9}$$

where $w_{b(\Delta t)}$ is a learnable bucketed scalar (log-bucketed lags), $\alpha, \beta > 0$ are learnable decay parameters for exponential attenuation, and $(a_p, b_p)$ are Fourier coefficients capturing periodic patterns (e.g., $p = 288$ for daily, $p = 144$ for half-daily periodicities at 5-minute resolution). This design distinguishes short vs. long lags, imposes monotonic decay, and embeds diurnal seasonality while keeping computation negligible.

**Graph-evolution-aware aggregation.** Spatial dependencies in traffic networks evolve over time due to demand shifts and external events, which are often overlooked by content-only attention. Given a node embedding $\mathbf{h}_i^{(t)} \in \mathbb{R}^{d_h}$, we obtain query, key, and value vectors via learnable projections: $\mathbf{q}_i^{(t)} = \mathbf{W}_Q \mathbf{h}_i^{(t)}$, $\mathbf{k}_i^{(t)} = \mathbf{W}_K \mathbf{h}_i^{(t)}$, $\mathbf{v}_i^{(t)} = \mathbf{W}_V \mathbf{h}_i^{(t)}$, with $\mathbf{W}_Q, \mathbf{W}_K, \mathbf{W}_V \in \mathbb{R}^{d_h \times d}$. To incorporate evolving structure, the dynamic adjacency $\hat{\mathbf{A}}_t$ is injected into the attention weights:

$$\alpha_{ij}^{(t)} = \text{softmax}_j \left( \frac{\mathbf{q}_i^{(t)} \cdot \mathbf{k}_j^{(t)}}{\sqrt{d}} + \gamma \log\left(\epsilon + \hat{\mathbf{A}}_t(i,j)\right) \right), \qquad \mathbf{h}_i^{(t+1)} = \sigma\left( \sum_{j=1}^{N} \alpha_{ij}^{(t)} \mathbf{v}_j^{(t)} \right). \tag{10}$$

This design fuses feature similarity with structural priors, enabling representations that adapt to graph evolution. The attention weight $\alpha_{ij}^{(t)}$ combines a content similarity score $\frac{\mathbf{q}_i^{(t)} \cdot \mathbf{k}_j^{(t)}}{\sqrt{d}}$, which measures the semantic correlation between nodes $i$ and $j$ at time $t$, with a log-adjusted adjacency prior $\log(\epsilon + \hat{\mathbf{A}}_t(i,j))$ scaled by a tunable coefficient $\gamma$. This design encourages the model to attend more to neighbors with strong structural correlation in $\hat{\mathbf{A}}_t$, while still preserving flexibility through content-based attention. The aggregated representation $\mathbf{h}_i^t$ therefore captures both feature-driven and structure-aware signals, enabling more faithful modeling of graph evolution in dynamic traffic flow forecasting.

## 4.2 GRAPH EVOLUTION-AWARE MASKED PRE-TRAINING

**Patch Embedding with Spatiotemporal Structure Preservation** Given an input sequence $\mathbf{X}'_{t:t+k-1} \in \mathbb{R}^{N \times K \times C}$, we employ hierarchical 2D convolutions to extract multi-scale representations:

$$\mathbf{X}_p^{(l)} = \text{Conv2D}^{(l)}(\mathbf{X}') \in \mathbb{R}^{N \times P_l \times d_l}, \quad l = 1, \dots, L, \tag{11}$$

where $P_l$ is the number of patches at level $l$ and $d_l$ the embedding dimension. To preserve spatiotemporal positions, we augment embeddings as

$$\mathbf{Z}_p^{(l)} = \mathbf{X}_p^{(l)} + \mathbf{P}_{\text{spatial}}^{(l)} \otimes \mathbf{P}_{\text{temporal}}^{(l)} + \mathbf{E}_{\text{scale}}^{(l)}, \tag{12}$$

where $\mathbf{P}_{\text{spatial}}^{(l)} \in \mathbb{R}^{N \times d_l}$ and $\mathbf{P}_{\text{temporal}}^{(l)} \in \mathbb{R}^{P_l \times d_l}$ encode positional priors, and $\mathbf{E}_{\text{scale}}^{(l)}$ captures hierarchical level.

**Adaptive Masking Strategy with Graph-Aware Constraints** Instead of uniform masking, we design a graph-aware stochastic masking scheme that incorporates structural priors:

$$\mathbf{M}_{ij} \sim \text{Bernoulli}\big(\rho \cdot \exp\big(-\gamma\,\mathcal{D}(i,j)\big)\big), \tag{13}$$

where $\mathcal{D}(i,j)$ denotes graph-based distance, $\gamma$ controls decay, and $\rho$ is the base mask ratio (scheduled by curriculum $r_s$). This encourages masking tokens from structurally correlated regions, forcing the model to leverage graph evolution for reconstruction.

### 4.2.1 REGULARIZED GRAPH EVOLUTION OBJECTIVE

To prevent the learned dynamic adjacency $\hat{\mathbf{A}}_t$ from overfitting, we constrain it with multiple regularizers:

$$\mathcal{L}_{\text{graph}} = \lambda_{\text{evo}}\mathcal{L}_{\text{evo}} + \lambda_{\text{sp}}\|\hat{\mathbf{A}}_t\|_1 + \lambda_{\text{str}}\|\hat{\mathbf{A}}_t - \mathbf{A}^{\text{static}}\|_F^2, \tag{14}$$

where $\mathcal{L}_{\text{evo}}$ enforces temporal smoothness across $\hat{\mathbf{A}}_t$, the $\ell_1$ term encourages sparsity, and the Frobenius penalty anchors the dynamic graph to a static prior $\mathbf{A}^{\text{static}}$.

**Pre-training Loss with Structural Alignment** The overall pre-training loss consists of reconstruction, structural alignment, and regularization terms. For each masked window $\{\mathbf{X}_t\}_{t=\tau}^{\tau+k-1}$, with masks $\{\mathbf{M}_t\}$, we reconstruct hidden tokens:

$$\mathcal{L}_{\text{rec}} = \sum_{t=\tau}^{\tau+k-1} \big\|(1 - \mathbf{M}_t) \odot \mathbf{X}_t - \hat{\mathbf{X}}_t\big\|_1, \tag{15}$$

$$\mathcal{L}_{\text{str}} = \sum_{t=\tau}^{\tau+k-1} \big\|\hat{\mathbf{A}}_t - \text{StopGrad}\big(\mathbf{G}_t^{\text{pseudo}}\big)\big\|_F^2, \quad \mathbf{G}_t^{\text{pseudo}}(i,j) \propto \exp\Big(-\frac{\|\mathbf{x}_{i,t}-\mathbf{x}_{j,t}\|_2^2}{\tau_g}\Big), \tag{16}$$

where $\mathbf{G}_t^{\text{pseudo}}$ is a pseudo-affinity graph computed only from *visible* tokens to avoid leakage.

The final pre-training objective is:

$$\mathcal{L}_{\text{pre}} = \mathcal{L}_{\text{rec}} + \lambda_{\text{str}}\mathcal{L}_{\text{str}} + \lambda_{\text{sp}}\sum_t \|\hat{\mathbf{A}}_t\|_1 + \lambda_{\text{tv}}\sum_t \|\hat{\mathbf{A}}_t - \hat{\mathbf{A}}_{t-1}\|_F^2. \tag{17}$$

This objective enforces that reconstructions rely on *evolving* spatial structures rather than static correlations, while regularization ensures sparsity, smoothness, and structural alignment.

### 4.3 UNIFIED FINE-TUNING PARADIGM

GEMFlow enables seamless integration with diverse spatiotemporal predictors by providing graph-evolution-aware representations that complement downstream hidden states. Specifically, we adopt GWNet Wu et al. (2019) as a representative backbone due to its strong performance, while also validating generality on DCRNN Li et al. (2018), MTGNN Wu et al. (2020), STID Shao et al. (2022), and STAEformer Liu et al. (2023). During fine-tuning, we input long-range sequences of $T_{\text{long}}$ steps into the pre-trained GEMFlow encoder, obtaining the representation $\hat{\mathbf{Z}}$. To align with the forecasting horizon, the last $P$ patches are truncated and reshaped into $\mathbf{Z}' \in \mathbb{R}^{N \times kD}$. These graph-evolution-aware embeddings are fused with the hidden states of the predictor (e.g., GWNet) through a lightweight MLP, producing the prediction representation $\hat{\mathcal{Y}} \in \mathbb{R}^{T_f \times N \times C}$. Finally, given the ground truth $\mathcal{Y} \in \mathbb{R}^{T_f \times N \times C}$, the regression objective is defined as:

$$\mathcal{L}_{\text{regression}} = \frac{1}{T_f N C} \sum_{j=1}^{T_f} \sum_{i=1}^{N} \sum_{k=1}^{C} \big|\hat{\mathcal{y}}_{ijk} - \mathcal{Y}_{ijk}\big|, \tag{18}$$

which enforces accurate forecasting while verifying that GEMFlow representations can universally enhance heterogeneous downstream architectures.

## 5 EXPERIMENT

### 5.1 EXPERIMENTAL SETUPS

**Datasets.**

To evaluate the effectiveness of our method, we conducted experiments on four public datasets from the Performance Measurement System (PeMS) Cao et al. (2020). PEMS03, PEMS04, PEMS07, and PEMS08 are traffic flow datasets collected by CalTrans PeMS Song et al. (2020).

The statistical information is summarized in Table 3. The datasets cover different geographical areas and time periods, providing a comprehensive testbed for evaluating spatiotemporal forecasting models. PEMS03 contains data from 358 sensors over 26,208 time steps, while PEMS04 includes 307 sensors with 16,992 observations. PEMS07 is the largest dataset with 883 sensors and 28,224 time steps, and PEMS08 consists of 170 sensors with 17,856

Figure 3: Statistics of datasets.

| Datasets | #Sensors | #Time Steps | #Time Interval |
|----------|----------|-------------|----------------|
| PEMS03   | 358      | 26208       | 5min           |
| PEMS04   | 307      | 16992       | 5min           |
| PEMS07   | 883      | 28224       | 5min           |
| PEMS08   | 170      | 17856       | 5min           |

recordings. All datasets use a 5-minute time interval, ensuring consistent temporal resolution across experiments.

This diverse set of datasets allows us to thoroughly assess the scalability and generalization capability of our proposed method under various conditions, including different network sizes, temporal durations, and geographical distributions. **Baselines.** We compare GEMFlow with baselines, including Transformer Vaswani et al. (2017), DCRNN Li et al. (2018), STGCN Yu et al. (2018), ASTGCN Guo et al. (2019), GWNet Wu et al. (2019), STSGCN Song et al. (2020), STFGNN Li & Zhu (2021), STGODE Fang et al. (2021), DSTAGNN Yu et al. (2018), ST-WA Cirstea et al. (2022), ASTGNN Guo et al. (2022), EnhanceNet Cirstea et al. (2021), AGCRN Bai et al. (2020), Z-GCNETs Chen et al. (2021), STEP Shao et al. (2022), PDFormer Jiang et al. (2023a), STAEformer Liu et al. (2023) and STD-MAE Gao et al. (2024).

**Settings.** We divide the PEMS03, PEMS04, PEMS07, and PEMS08 datasets into training, validation, and test sets according to a 6:2:2 ratio according to the previous baselines Song et al. (2020); Li & Zhu (2021); Fang et al. (2021); Jiang et al. (2023a); Guo et al. (2019). In the pre-training phase, we followed the setup Gao et al. (2024) and set the long time series of the four datasets as 864 time steps, respectively. The output length of forcasting is 12. The encoder have 2 GNN layers and 4 transformer layers while the decoder have 2 GNN layers and 1 transformer layer. The number of multi-attention heads in transformer layer is set to 4. We use a patch size $L$ of 12 to align with the forecasting input.The hidden dimension of the latent representations of GNNs and TSFormer $d$ is set to 96 and 32, respectively. The masking ratio $r$ is set to 0.25. The loss function is mean absolute error (MAE). For evaluation, we use MAE, root mean squared error (RMSE), and mean absolute percentage error (MAPE(%)). (Our environment: CPU: Intel(R) Xeon(R) Silver 4210 CPU @ 2.20GHz, GPU: NVIDIA RTX 4090@24GB, Memory: 128GB. The implementation of our model and all baselines are based on Pytorch 1.9.0 and Python 3.9)

### 5.2 MAIN RESULTS

Across the PEMS03, PEMS04, PEMS07, and PEMS08 datasets, our proposed GEMFlow framework consistently achieves superior performance over a range of state-of-the-art baseline models, as presented in Table 1. The baseline results are directly taken from the original literature, which is widely cited and recognized within the spatiotemporal forecasting community, thereby ensuring the fairness and credibility of the comparison.GEMFlow demonstrates substantial improvements across all evaluation metrics, including MAE, RMSE, and MAPE, underscoring its robustness and generalization capability in modeling complex spatiotemporal dependencies. The consistent performance gains affirm GEMFlow's ability to capture both spatial and temporal structures more effectively than traditional methods. Notably, models that integrate spatiotemporal representations—such as GEMFlow—consistently outperform conventional time series models, owing to their enhanced capacity to learn intricate interactions across both spatial and temporal dimensions.

Table 1: Performance comparison on four datasets.

| Model | PEMS03 | | | PEMS04 | | | PEMS07 | | | PEMS08 | | |
|---|---|---|---|---|---|---|---|---|---|---|---|---|
| | MAE | RMSE | MAPE | MAE | RMSE | MAPE | MAE | RMSE | MAPE | MAE | RMSE | MAPE |
| Transformer | 17.50 | 30.24 | 16.80 | 23.83 | 37.19 | 15.57 | 26.80 | 42.95 | 12.11 | 18.52 | 28.68 | 13.66 |
| DCRNN | 18.18 | 30.31 | 18.91 | 24.70 | 38.12 | 17.12 | 25.30 | 38.58 | 11.66 | 17.86 | 27.83 | 11.45 |
| STGCN | 17.49 | 30.12 | 17.15 | 22.70 | 35.55 | 14.59 | 25.38 | 38.78 | 11.08 | 18.02 | 27.83 | 11.40 |
| ASTGCN | 17.69 | 29.66 | 19.40 | 22.93 | 35.22 | 16.56 | 28.05 | 42.57 | 13.92 | 18.61 | 28.16 | 13.08 |
| GWNet | 19.85 | 32.94 | 19.31 | 25.45 | 39.70 | 17.29 | 26.85 | 42.78 | 12.12 | 19.13 | 31.05 | 12.68 |
| STSGCN | 17.48 | 29.21 | 16.78 | 21.19 | 33.65 | 13.90 | 24.26 | 39.03 | 10.21 | 17.13 | 26.80 | 10.96 |
| STFGNN | 16.77 | 28.34 | 16.30 | 19.83 | 31.88 | 13.02 | 22.07 | 35.80 | 9.21 | 16.64 | 26.22 | 10.60 |
| STGODE | 16.50 | 27.84 | 16.69 | 20.84 | 32.82 | 13.77 | 22.99 | 37.54 | 10.14 | 16.81 | 25.97 | 10.62 |
| DSTAGNN | 15.57 | 27.21 | 14.68 | 19.30 | 31.46 | 12.70 | 21.42 | 34.51 | 9.01 | 15.67 | 24.77 | 9.94 |
| ST-WA | 15.17 | 26.63 | 15.83 | 19.06 | 31.02 | 12.52 | 20.74 | 34.05 | 8.77 | 15.41 | 24.62 | 9.94 |
| ASTGNN | 15.07 | 26.88 | 15.80 | 19.26 | 31.16 | 12.65 | 22.23 | 35.95 | 9.25 | 15.98 | 25.67 | 9.97 |
| EnhanceNet | 16.05 | 28.33 | 15.83 | 20.44 | 32.37 | 13.58 | 21.87 | 35.57 | 9.13 | 16.33 | 25.46 | 10.39 |
| AGCRN | 16.06 | 28.49 | 15.85 | 19.83 | 32.26 | 12.97 | 21.29 | 35.12 | 8.97 | 15.95 | 25.22 | 10.09 |
| Z-GCNETs | 16.64 | 28.15 | 16.39 | 19.50 | 31.61 | 12.78 | 21.77 | 35.17 | 9.25 | 15.76 | 25.11 | 10.01 |
| STNorm | 15.32 | 25.93 | 14.37 | 19.21 | 32.30 | 13.05 | 20.59 | 34.86 | 8.61 | 15.39 | 24.80 | 9.91 |
| STEP | 14.22 | 24.55 | 14.42 | 18.20 | 29.71 | 12.48 | 19.32 | 32.19 | 8.12 | 14.00 | 23.41 | 9.50 |
| PDFormer | 14.94 | 25.39 | 15.82 | 18.32 | 29.97 | 12.10 | 19.83 | 32.87 | 8.53 | 13.58 | 23.51 | 9.05 |
| STAEformer | 15.35 | 27.55 | 15.18 | 18.22 | 30.18 | 11.98 | 19.14 | 32.60 | 8.01 | 13.46 | 23.25 | 8.88 |
| STD-MAE | 13.80 | 24.43 | 13.96 | 17.80 | 29.25 | 11.97 | 18.65 | 31.44 | 7.84 | 13.44 | 22.47 | 8.76 |
| **GEMFlow** | **13.61** | **23.91** | **13.10** | **16.73** | **28.39** | **11.10** | **17.78** | **30.78** | **7.24** | **12.97** | **22.13** | **7.91** |

## 5.3 ABLATION STUDY

Table 2: Performance drop (↑: worse performance) of framework ablation.

| Variants | Metric | PEMS03 | PEMS04 | PEMS07 | PEMS08 |
|---|---|---|---|---|---|
| NoDAM | MAE | +1.37 | +1.48 | +1.33 | +1.13 |
| | RMSE | +1.20 | +1.52 | +1.33 | +1.74 |
| | MAPE | +1.77 | +2.07 | +1.75 | +1.34 |
| NoGNN | MAE | +1.50 | +2.65 | +1.12 | +1.15 |
| | RMSE | +2.81 | +2.42 | +3.07 | +2.59 |
| | MAPE | +2.21 | +4.83 | +3.24 | +2.47 |
| NoM | MAE | +2.01 | +1.66 | +1.01 | +1.40 |
| | RMSE | +1.96 | +2.54 | +2.50 | +2.68 |
| | MAPE | +2.81 | +3.77 | +2.93 | +2.40 |

**Framework Ablation.** To assess the contribution of each component in GEMFlow, we design three ablated variants. (1) NoDAM. Without utilizing dyanamic adjacency matrix. (2) NoGNN. Replaceing GNN module with MLP. (3) NoM. Without applying masking during spatiotemporal pre-training. As shown in Table 2, removing the dynamic adjacency matrix results in a clear performance drop, demonstrating the importance of modeling evolving spatial dependencies. Replacing the GNN with an MLP leads to further degradation, confirming the GNN's strength in capturing complex spatiotemporal structures. Without the masking mechanism, the model fails to learn rich representations, significantly weakening downstream performance. Overall, GEMFlow achieves the best results when all components are jointly applied, validating the complementary roles of dynamic graph modeling, GNN-based representation learning, and masked pre-training.

## 5.4 HYPER-PARAMETER ANALYSIS

In our hyper-parameter study, we investigated the impact of pre-training length $k$ and mask ratio on GEMFlow's performance. As shown in Figure 4, setting $k = 864$ yields the best results, offering a good trade-off between contextual richness and training cost. This length provides sufficient temporal scope to capture complex spatiotemporal patterns. Importantly, GEMFlow remains robust to variations in $k$, maintaining strong performance even with shorter or longer sequences.

We also evaluated different mask ratios, as shown in Figure 5. A ratio of 0.25 was found optimal—masking enough information to encourage meaningful representation learning, without overly

Figure 4: Hyper-parameter study on pre-training length $k$

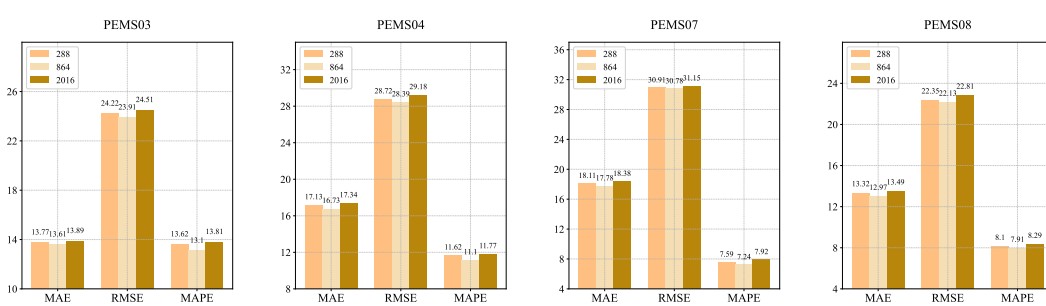

disrupting the input structure. Again, GEMFlow shows low sensitivity across a range of ratios, underscoring its stability and reducing reliance on intensive tuning.

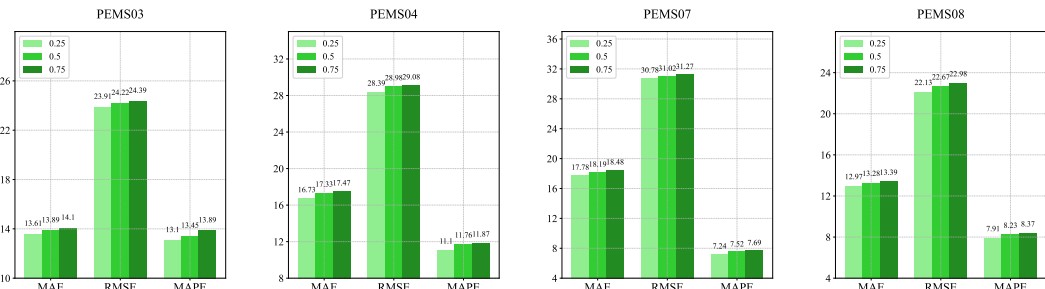

Figure 5: Hyper-parameter study on mask ratio $r$

## 5.5 EFFICIENCY ANALYSIS

As shown in Table 3, GEMFlow significantly reduces computational time compared to other pre-training models (STEP and STD-MAE) across all datasets. This efficiency stems from GEMFlow's adaptive graph neural network and optimized pre-training strategy, which enable faster convergence and more efficient resource use. Achieving superior speed while maintaining high accuracy makes GEMFlow particularly suitable for large-scale spatiotemporal forecasting tasks.

Table 3: Efficiency comparison with pre-training models.

| Datasets | STEP | STD-MAE | GEMFlow |
|---|---|---|---|
| PEMS03 | 108ms | 50ms | **31ms** |
| PEMS04 | 73ms | 34ms | **27ms** |
| PEMS07 | 516ms | 142ms | **99ms** |
| PEMS08 | 62ms | 48ms | **35ms** |

## 6 CONCLUSION

This paper presents GEMFlow, an Adaptive Graph Neural Network enhanced pre-training framework tailored for spatiotemporal forecasting. By integrating dynamic adjacency matrices with transformer-based architectures, GEMFlow effectively captures complex spatial and temporal dependencies inherent in spatiotemporal data. Extensive experiments on multiple benchmark datasets demonstrate that GEMFlow consistently outperforms existing methods, showcasing strong robustness and generalization across diverse downstream tasks. Our framework not only advances the state of the art but also establishes a new benchmark for versatile and efficient spatiotemporal modeling.

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

## A  APPENDIX

You may include other additional sections here.

