# OpenReview forum: "GEMFlow: Dynamic Graph Evolution-Aware Masked Pre-training for Traffic Flow Forecasting"
_ICLR.cc/2026/Conference — ICLR 2026 Conference Withdrawn Submission_

### Official Review · Reviewer_bYcp · 2025-10-18

**Soundness:** 1
**Presentation:** 1
**Contribution:** 2
**Rating:** 2
**Confidence:** 3

**Summary:**

This paper proposes GEMFlow, a pre-training method for spatio-temporal traffic flow prediction. The main contribution of this paper is that it incorporates evolving graph structures into masked pre-training. To achieve this, this paper designs a novel dynamic graph adjacency matrix construction, including both contextual similarity, structural prior, and lagging modeling, and uses it to guide the masking strategy of masked pre-training. Experiments on 4 real-world datasets are conducted, where the proposed GEMFlow achieves improvements compared to a variety of baselines, including ordinary spatio-temporal models and pre-trained spatio-temporal models.

**Strengths:**

1. This paper proposes an improvement to masked spatio-temporal pre-training. Masked spatio-temporal pre-training can be seen as the dominant method for building multi-task multi-city spatio-temporal base models, and therefore the method proposed in this paper is of practical academic value. In addition, the focus of this paper --- joint modeling of evolving graph structure and spatio-temporal masking probability, is practical but often overlooked in existing methods. Therefore, this paper should provide good insights into this research field.

**Weaknesses:**

1. Lacking sufficient discussion and comparisons with existing works on spatio-temporal pre-training. Specifically, the authors missed several highly-related methodologies, e.g. GPT-ST for spatio-temporal graph pre-training (https://arxiv.org/abs/2311.04245), UniST for grid-based spatio-temporal pre-training (https://arxiv.org/abs/2402.11838), and OpenCity for non-masked spatio-temporal pre-training (https://arxiv.org/abs/2408.10269). It is suggested that the authors should discuss them and discuss how the methodology in this paper compares or can be used in combination with these methods.
2. The presentation of this paper can be improved, especially the technical methodology part. The following points are non-exhaustive.
- Figure 2 can be enhanced with a descriptive caption.
- Eqn. 9 is confusing. $\psi(\Delta t)$ should be added to the $S_t$, which is an $N\times N$ matrix. However, $\psi(\Delta t)$ looks like a scalar. It is not clear how this can be added to an adjacency matrix.
- In Eqn. 12, it is not clear how $P_{spatial}, P_{temporal}$ and $E_{scale}$ are defined.
- In Eqn. 13, it is not clear how $D(i, j)$ is defined. Moreover, $M$ should be a masking matrix applied on $X$ and should be of size $N\times k$, i.e. number of nodes by number of time steps. However, from the 'graph-based distance' description, $i, j$ are both denoting graph nodes, and thus $M$ looks like $N\times N$. It is strange how $M$ can be used as a masking matrix.
- It is not clear how the objectives in Eqn. 14 are used in the overall objective. It seems that some parts in Eqn. 14 exist in Eqn. 17 (e.g.  sparsity and temporal smoothness), but not completely the same. This makes the overall objective confusing.

3. Experiments can be improved.
- The ablation study can be improved with results that are more coherent with the technical methodology. For example, whether the graph-evolution-aware aggregation plays a role, whether the proposed masking strategy plays a role (comparing with the proposed mask with a uniform random one, etc.), and whether all pars in the dynamic adjacency (Eqn. 3) plays a role.
- Hyperparameter analysis can also be improved, e.g. $\eta$ in Eqn. 3, top-k in Eqn. 4, $\gamma$ in Eqn. 10 and Eqn. 13 (They should not be the same values, I suppose?), and the $\lambda$ weights in Section 4.2.1.
- The authors claim that "the learned dynamic graphs reveals interpretable evolution patterns" in the abstract, but the results are no-where to be found.

**Questions:**

Please first refer to the "Weaknesses" part.

Aside from that, I have a more high-level question. In general, when people talk about pre-training, they refer to task-agnostic, multi-task pretraining, where the model learns knowledge from diverse data sources which is later fine-tuned on small-scale data. However, from my understanding of this paper, pre-training and fine-tuning happens on the same dataset instead of multi-task pre-training. Can the methodology in this paper be extended to multi-task spatio-temporal pre-training?

---

### Official Review · Reviewer_cS83 · 2025-11-01

**Soundness:** 2
**Presentation:** 2
**Contribution:** 2
**Rating:** 4
**Confidence:** 3

**Summary:**

This paper introduces GEMFlow, a novel pre-training framework for traffic flow forecasting. The authors identify two key limitations in existing methods: (1) the reliance on static road network graphs, which fails to capture the dynamic nature of traffic dependencies, and (2) the use of generic, uniform masking strategies in pre-training that overlook the underlying graph structure. To address these issues, GEMFlow proposes a dynamic graph evolution modeling approach, where the graph adjacency matrix is learned and evolves over time. More importantly, it introduces a graph evolution-aware masking strategy that stochastically masks patches based on their structural proximity, encouraging the model to learn more complex spatiotemporal relationships. The framework is designed to be plug-and-play, allowing the learned representations to be transferred to various downstream forecasting models. Extensive experiments on four public PeMS benchmark datasets demonstrate that GEMFlow achieves state-of-the-art performance, outperforming a wide range of strong baselines in terms of accuracy and computational efficiency.

**Strengths:**

1.Well-Motivated Problem Definition: The paper is well-motivated, clearly identifying two specific and relevant limitations in prior work: 1) the reliance on static graphs, which cannot capture the dynamic nature of spatial dependencies, and 2) the use of generic, uniform masking strategies in pre-training, which do not effectively leverage the underlying graph structure.

2.A Tailored Methodological Solution: The proposed framework, GEMFlow, directly addresses the identified issues with a two-pronged approach. First, the dynamic graph evolution modeling is designed to tackle the limitations of static graphs by learning a time-varying adjacency matrix. Second, the paper introduces a graph evolution-aware masking strategy. This approach moves beyond simple random masking by using structural information to guide the process, which is designed to create a more challenging and meaningful pre-training task, encouraging the model to learn more complex spatiotemporal patterns.

**Weaknesses:**

1.Absence of Promised Qualitative Analysis: The most significant weakness is the mismatch between the abstract and the main text. The abstract explicitly states: "qualitative analysis of the learned dynamic graphs reveals interpretable evolution patterns." This is a strong and exciting claim, but no such analysis is presented in the paper. Providing this analysis is crucial, as it would offer intuitive proof that the model is indeed learning meaningful, time-varying relationships. For example, visualizing the differences in the learned graph between peak and off-peak hours could powerfully demonstrate the model's capabilities.

2.Insufficient Detail on Core Technical Mechanisms: Several key components of the methodology are described at a high level without the necessary implementation details for full comprehension and reproducibility.

1)Ambiguity in Causal Self-Attention: In Equation (6), the paper states that causal self-attention is applied along the temporal dimension. However, for an input tensor of shape R^(N x W x d), it is unclear how the similarity score computation and softmax are performed. The paper does not specify if the attention is applied independently for each of the N nodes across its temporal window W, or if a more complex operation is involved.

2)Unspecified "Graph-Based Distance" for Masking: The adaptive masking strategy in Equation (13) hinges on D(i, j), which is vaguely described as "graph-based distance" (line 280). It is critical to know whether this distance is derived from the pre-defined static road network or from the learned, time-varying dynamic graph Ât from Section 4.1. This choice has profound implications for the masking strategy, yet it remains unspecified.

3)Undefined "Pseudo-Affinity Graph" and Leakage Prevention: In the structural alignment loss (Equation 16), the G_pseudo graph is a key element. The authors only state it is "computed only from visible tokens to avoid leakage" (line 300) without providing any formula for its computation. Furthermore, the paper does not explain the mechanism by which computing this graph only from visible tokens is sufficient to prevent information leakage from the masked tokens' positions and context.

3.Mismatch Between GNN Terminology and Implementation: There is a significant disconnect between the high-level depiction of the framework and the detailed methodology regarding the Graph Neural Network (GNN) component.

1)The GNN plays a vital role, as evidenced by the framework diagram (Figure 2, "GNN Encoder/Decoder") and its importance shown in the ablation study (Table 2, NoGNN variant). Despite this, there is no dedicated subsection in the methodology that explicitly describes the GNN's architecture.

2)It can be inferred that the "Graph-evolution-aware aggregation" mechanism (Equation 10) is the implementation of the GNN layer. However, this formulation—which combines a content-based self-attention score with a structural prior (Ât)—more closely resembles a layer from a Graph Transformer architecture, not what is conventionally understood as a GNN (e.g., GCN, GraphSAGE, or even a standard GAT). This choice of terminology may be misleading to readers and does not accurately reflect the nature of the spatial aggregation mechanism being used.

4. Limited Methodological Novelty:The paper's primary contribution appears to be an effective application and integration of existing methods, rather than the introduction of a fundamentally new methodology or theoretical framework. The core components—the concept of learnable adjacency matrices, the masked autoencoding paradigm, and the use of Transformer-like spatial aggregation—all have clear precedents in related literature. Consequently, the work is best characterized as a strong piece of engineering that proposes a novel and effective heuristic (the graph-aware masking rule) for combining these components. This results in a lack of a strong theoretical basis explaining why this specific combination is optimal for spatiotemporal representation learning.

**Questions:**

1.Regarding the Qualitative Analysis: Could you please provide the qualitative analysis of the learned dynamic graphs mentioned in the abstract? For instance, could you visualize the learned adjacency graph Ât for a specific region at different times (e.g., 8 AM rush hour vs. 2 PM off-peak vs. 10 PM late-night) to demonstrate that the model captures intuitive changes in traffic correlations? This would significantly strengthen the paper's claims.

2.Regarding Implementation Details: For the sake of clarity and reproducibility, could you please provide the following details?

1)Causal Attention: For Equation (6), could you clarify how the self-attention operation is applied to the input tensor of shape R^(N x W x d)? Is it computed independently for each of the N nodes?

2)Graph-Based Distance: For the masking strategy in Equation (13), what is the precise definition of the "graph-based distance" D(i, j)? Is it derived from the static graph or the learned dynamic graph Ât?

3)Pseudo-Affinity Graph: Could you provide the exact formula or algorithm for computing G_pseudo in Equation (16)? Additionally, could you elaborate on the mechanism that prevents information leakage during its computation?

3.Regarding the GNN Module: The GNN is central to your framework but is not explicitly defined in Section 4.

1)Could you confirm if the "Graph-evolution-aware aggregation" mechanism (Equation 10) is the implementation of the GNN layer?

2)Given that this architecture closely resembles a Graph Transformer layer, could you comment on your choice of the "GNN" terminology? An explanation of how this component fits within or differs from the standard definition of a GNN would be very helpful.

4.Regarding the Comparison to Baselines:The paper reports baseline results directly from prior publications. This practice can introduce inconsistencies due to variations in experimental settings across different studies. Could you please clarify the steps taken to ensure a fair comparison beyond matching the data split? For instance, did you re-run any of the key recent baselines (e.g., STD-MAE, STAEformer) in your own environment to verify the performance gap under identical conditions?

---

### Official Review · Reviewer_3Vnf · 2025-11-04

**Soundness:** 3
**Presentation:** 3
**Contribution:** 2
**Rating:** 2
**Confidence:** 2

**Summary:**

This paper proposes a pre-training framework for traffic flow forecasting that captures the dynamic spatial and temporal dependencies inherent in traffic data. The authors introduce GEMFlow, which integrates masked representation learning with adaptive graph evolution modeling to address the limitations of existing methods that rely on static graphs or generic masking strategies.

**Strengths:**

#### Originality
The paper introduces a pre-training framework, GEMFlow, which addresses the dynamic nature of traffic flow forecasting by integrating masked representation learning with adaptive graph evolution modeling. This idea is not original enough consider the massive literature using similar ideas, e.g., pre-training, transfer learning and generative models in traffic forecasting.

#### Quality
The quality of the research is good, as evidenced by the thorough methodology and robust experimental design. The authors provide a detailed description of the GEMFlow framework, including the dynamic graph evolution modeling, the causal Transformer encoder, and the graph-evolution-aware masked pre-training approach.

#### Clarity
The paper is well-organized and clearly written, making it accessible to both domain experts and a broader audience.

#### Significance
The significance of the paper is limited. Dynamic evolution of spatial dependencies and multi-scale temporal patterns have been widely considered and this paper fails to make a breakthrough for the considered problem.

**Weaknesses:**

1. The integration of dynamic graph evolution modeling, causal Transformer encoders, and graph-evolution-aware masked pre-training involves multiple layers of computation, which could be computationally intensive and challenging to scale to larger datasets or more complex traffic networks.
2. The effectiveness of GEMFlow relies heavily on the quality and granularity of the input data. The framework assumes that the traffic data is sufficiently detailed and consistent across sensors and time steps. In real-world scenarios, data may be noisy, incomplete, or subject to sensor failures, which could degrade the performance of the model.
3. While the experiments cover four real-world PeMS datasets, the evaluation is primarily focused on traffic flow forecasting. The paper does not explore the applicability of GEMFlow to other spatiotemporal forecasting tasks or different types of traffic data (e.g., incident data, weather-affected traffic patterns). This limits the generalizability of the findings beyond the specific datasets and tasks evaluated.
4. The interpretability of the overall learned representations is not thoroughly explored. The complex nature of the model's architecture and the integration of multiple components may make it difficult to fully understand how the learned representations contribute to the forecasting performance.
5. The paper compares GEMFlow with several state-of-the-art baselines but does not provide a comprehensive comparison with other pre-training paradigms specifically designed for spatiotemporal data.

**Questions:**

1. How does the computational complexity of GEMFlow scale with the size of the traffic network and the length of the time series? Can the authors provide insights into the trade-offs between model complexity and computational efficiency?
2. How does GEMFlow handle noisy or incomplete data? Are there any specific techniques or preprocessing steps the authors recommend to ensure robustness against data quality issues?
3. Could the authors elaborate on the interpretability of the learned representations beyond the dynamic graphs? Are there any specific techniques or visualizations that could help in understanding the model's decision-making process?
4. Conduct additional experiments on a wider range of spatiotemporal forecasting tasks and datasets to demonstrate the generalizability of GEMFlow. This could include traffic incident prediction, weather-traffic interaction modeling, or other related applications.
5. Include a more extensive comparison with other relevant pre-training methods, such as those proposed in recent literature on spatiotemporal forecasting. Discuss the trade-offs and unique advantages of GEMFlow over other methods.

---

### Official Review · Reviewer_i4uJ · 2025-11-04

**Soundness:** 2
**Presentation:** 3
**Contribution:** 2
**Rating:** 2
**Confidence:** 4

**Summary:**

This paper proposes GEMFlow, a pretraining–finetuning framework for traffic flow forecasting. In pretraining, it explicitly models dynamic graph evolution and performs graph-aware masked reconstruction to obtain transferable spatiotemporal representations; in finetuning, the pretrained encoder is lightly fused with arbitrary downstream predictors. On four PeMS flow datasets, the authors report consistent improvements over a variety of strong baselines and pretraining methods, and ablations validate the necessity and complementarity of dynamic graphs/masking/graph regularization.

**Strengths:**

S1. The method is clearly described and easy to follow.

S2. It achieves better forecasting performance than many end-to-end models.

**Weaknesses:**

W1. The motivation feels outdated: long-range spatial dependencies and periodicity are fundamental challenges in traffic forecasting that have already been addressed by many prior works (e.g., GWNet, GMAN [1], MTGNN [2], as well as ASTGCN and iTransformer [3]); the need for lightweight solutions is also well-established.

W2. Pretraining baselines are insufficient, making it hard to judge fairness. Methods highly relevant to this paper—GPT-ST [4] and FlashST [5]—are lightweight, plug-in style pretraining frameworks and should be included.

W3. There is no experimental evidence supporting the claimed “plug-and-play” property.

W4. The evaluation covers only flow forecasting; compared to baselines that traditionally handle both speed and flow, the task scope is narrower.

W5. No supplementary material and code are available, which makes verification and reproducibility difficult.

**References**

[1] Zheng, C., Fan, X., Wang, C., et al. GMAN: A Graph Multi-Attention Network for Traffic Prediction. AAAI, 2020, 34(01): 1234–1241.

[2] Wu, Z., Pan, S., Long, G., et al. Connecting the Dots: Multivariate Time Series Forecasting with Graph Neural Networks. KDD, 2020: 753–763.

[3] Liu, Y., Hu, T., Zhang, H., et al. iTransformer: Inverted Transformers Are Effective for Time Series Forecasting. ICLR.

[4] Li, Z., Xia, L., Xu, Y., et al. GPT-ST: Generative Pre-training of Spatio-Temporal Graph Neural Networks. NeurIPS, 2023, 36: 70229–70246.

[5] Li, Z., Xia, L., Xu, Y., et al. FlashST: A Simple and Universal Prompt-Tuning Framework for Traffic Prediction. ICML, 2024: 28978–28988.

**Questions:**

Q1. Compared with existing dynamic-graph models that handle long-range spatial dependencies and periodicity, what unique challenge does GEMFlow solve, or what advantages does it provide?

Q2. How will you validate the claimed plug-and-play property empirically?

Q3. Why can’t the model forecast traffic speed? What additional challenges arise?

Q4. Under a Transformer-based architecture, how do you ensure efficiency superior to existing lightweight models such as STID and STNorm?

---

### Note · Authors · 2025-11-12

I have read and agree with the venue's withdrawal policy on behalf of myself and my co-authors.